 eLIFE

# Strong inter-population cooperation leads to partner intermixing in microbial communities

Babak Momeni[1]\*, Kristen A Brileya[2], Matthew W Fields[2], Wenying Shou[1]\*

[1]Division of Basic Sciences, Fred Hutchinson Cancer Research Center, Seattle, United States; [2]Department of Microbiology and Center for Biofilm Engineering, Montana State University, Bozeman, United States

**Abstract** Patterns of spatial positioning of individuals within microbial communities are often critical to community function. However, understanding patterning in natural communities is hampered by the multitude of cell–cell and cell–environment interactions as well as environmental variability. Here, through simulations and experiments on communities in defined environments, we examined how ecological interactions between two distinct partners impacted community patterning. We found that in strong cooperation with spatially localized large fitness benefits to both partners, a unique pattern is generated: partners spatially intermixed by appearing successively on top of each other, insensitive to initial conditions and interaction dynamics. Intermixing was experimentally observed in two obligatory cooperative systems: an engineered yeast community cooperating through metabolite-exchanges and a methane-producing community cooperating through redox-coupling. Even in simulated communities consisting of several species, most of the strongly-cooperating pairs appeared intermixed. Thus, when ecological interactions are the major patterning force, strong cooperation leads to partner intermixing.

**\*For correspondence:**
bmomeni@fhcrc.org (BM);
wenying.shou@gmail.com (WS)

**Competing interests:** The authors have declared that no competing interests exist

**Reviewing editor**: Diethard Tautz, Max Planck Institute for Evolutionary Biology, Germany

## Introduction

Biological interactions drive pattern formation at different levels of organization (*Murray, 2003*), ranging from developmental patterning within multicellular organisms and biofilms (*Shapiro, 1998*; *Lewis, 2008*; *Vlamakis et al., 2008*; *Chuong and Richardson, 2009*), to ecological patterning within multispecies communities (*Levin, 1992*; *Rietkerk and van de Koppel, 2008*; *Momeni et al., 2011*). Patterning, reflecting the relative spatial positioning of individuals with respect to each other, can be critical for the proper functioning of a community. Consider microbial communities: in a synthetic community, three bacterial species, each contributing an essential benefit while simultaneously competing for these benefits, can only grow when they are separated by an intermediate distance (*Kim et al., 2008*); different types of patterning are correlated with different levels of biofilm growth (*Christensen et al., 2002*; *Brenner and Arnold, 2011*); branching colony morphology allows more effective spreading across a nutrient-poor surface (*Levine and Ben-Jacob, 2004*); and in waste treatment granules, the layered pattern of bacteria and archaea is thought to facilitate the sequential degradation of substrates (*Satoh et al., 2007*).

Despite the wide-ranging importance of microbial communities in, for example, human health and the biogeochemical cycling of elements, it is still unclear how cell–cell and cell–environment interactions govern the patterning of communities (*Elias and Banin, 2012*). Understanding the mechanistic basis of pattern formation from observations of natural communities is stymied by the multitude of cell–cell and cell–environment interactions, as well as environmental variations within and across communities. Thus, it is not uncommon to observe qualitatively different patterns in samples of essentially the same type of community (*Christensen et al., 2002*; *Wilmes et al., 2008*; *Dekas et al., 2009*).

**eLife digest** Microorganisms such as bacteria, archaea and tiny eukaryotes are found throughout the biosphere. Some of these microorganisms are pathogens that cause diseases in animals, while others provide nutrients, including essential amino acids and vitamins; there are also microorganisms that have critical roles in recycling elements such as carbon, nitrogen and oxygen in the biosphere. In the natural world, microorganisms interact with their environment and with each other, often competing for space, light and nutrients, but sometimes they act cooperatively, which benefits all parties involved.

Microbial communities exhibit spatial patterns that reflect the relative positioning of different microbes in a community. These patterns can be critical for the proper functioning of a microbial community. For example, in the microbial granules that digest organic compounds in waste water, the stratified pattern of different microbial species can be thought of as a sequence of catalysts needed to perform a series of biochemical processing steps. Thus, it is important to understand the mechanisms that drive pattern formation in multispecies communities.

Now, through a combination of simulations and experiments, Momeni et al. have identified two features of spatial patterns in two-population microbial communities when pattern formation is driven by fitness effects related to the ecological interactions between cells. First, interactions that confer significant advantages to at least one of the populations can potentially result in the generation of a stable community; the community is stable in the sense that if it is disturbed, it will return to its stable population composition following the disturbance. Indeed, in engineered *Saccharomyces cerevisiae* communities, very different initial population ratios converged to the same value over time when one strain depended on the other strain, or when the two strains depended on each other, but not when the two strains competed.

The second feature applies to microbial communities composed of two cooperating populations: whereas two populations that compete with each other tend to segregate, cooperation results in the members of the two populations mixing together. Momeni et al. observe the formation of such an "intermixed" community in simulations, and also in two experimental systems that involve cooperation—a community containing two different strains of yeast cooperating through metabolite exchange, and a biofilm in which *Methanococcus maripaludis*, an archaeon that produces methane, cooperates with the bacterium *Desulfovibrio vulgaris*.

These two features of spatial patterning are conceptually similar to the competitive exclusion principle, which states that two species competing for the same resources cannot stably coexist if competition is the sole force at work. This principle has, therefore, encouraged scientists to search for the other forces that must be responsible for the coexistence of different species. Similarly, by predicting the sorts of patterns that will form when the fitness effects of ecological interactions between cells are the only forces at work, Momeni et al lay the groundwork for investigations into other mechanisms, such as cell–environment interactions and active cell motility, that can govern pattern formation in microbial communities.

To circumvent the lack of control in natural communities, we employed mathematical and experimental systems to systematically investigate how different types of ecological interactions might lead to distinct community patterning. Interactions can be classified into different ecological types based on their fitness effects on the interacting partners. We focused on the fitness effects rather than the molecular mechanisms of interactions, because diverse molecular mechanisms, ranging from physical associations in cell coaggregates and biofilms (*Kolenbrander et al., 2010*) to chemical interactions such as quorum sensing (*Parsek and Greenberg, 2005*), toxin warfare (*Vetsigian et al., 2011*), and metabolite supply (*Christensen et al., 2002*), all have fitness consequences which can be positive, neutral, or negative. Among different ecological interactions, we have placed a special emphasis on strong cooperation, interactions with large positive fitness effects on both partners including obligatory cooperation. This is because 1) it is important in a wide variety of microbial communities ranging from syntrophic systems critical for nutrient cycling (*Schink, 2002*; *Falkowski et al., 2008*; *McInerney et al., 2009*) to pathogenic biofilms (*Kelly, 1980*; *Kolenbrander et al., 2010*; *Elias and Banin, 2012*); and 2) the codependence

between cooperative partners poses special challenges for isolating and culturing cells (*Schink, 2002*; *McInerney et al., 2009*).

We investigated patterning in three-dimensional communities grown from two fluorescently-marked populations of cells initially randomly distributed on top of a surface (*Figure 1A*). Starting with a generalized model based on fitness effects of ecological interactions between two populations (A and B) occurring at a local scale ('fitness model'), we predicted: 1) interactions benefiting at least one partner could potentially allow initially disparate partner ratios to converge over time, and 2) unlike other types of ecological interactions that caused partner segregation or layering of one population over the other (A over B or B over A), strongly cooperating partners intermixed by forming patches that successively accumulated on top of each other (A over B over A over B, etc). We tested these predictions experimentally in obligatory cooperative systems including engineered yeast communities and syntrophic methanogenic biofilms. Finally, we used the fitness model to show 'strongly cooperating partners intermix' could be generalized to communities consisting of multiple species.

## Results

All microbial communities exhibit intra- and inter-population competition as cells compete for shared resources, including space. Therefore, for simplicity, we use [~ ~] to denote the 'baseline' competition. The fitness effect of baseline competition can start at zero if shared resources are in excess, but will eventually become negative as shared resources become limited. In 'addition' to competition, interactions between two partners can exert positive, negative, or no fitness effects on one or both partners. The 'net' fitness effects of all interactions between two partners, including competition, on the two partners can be represented as two symbols in a square bracket. There are six possibilities:[~ ~] (no fitness effects other than those from baseline competition), [~ ↑] (commensalism, in which one partner experiences nothing more than competition whereas the other enjoys an overall fitness benefit even when competition has been taken into consideration), [~ ↓] (amensalism), [↓ ↓] (mutual antagonism), [↓ ↑] (victim-exploiter), and [↑ ↑] (cooperation). The identities of partners may be added to the notation such that A[↓ ↑]B would mean that the overall interaction (including competition) inhibits A and promotes B. Under this notation, inter-population toxin-warfare (*Vetsigian et al., 2011*) would be [↓ ↓] while inter-population cooperation based on the exchange of distinct net benefits would be [↑ ↑]. Some interactions may at the first sight seem cooperative, but the net interaction may turn out not to be cooperative after considering the negative effects of competition (*Kim et al., 2008*). In the game theory definition, cooperative acts incur fitness costs to the performers. Here we have taken a more liberal definition of cooperation to include mutually-beneficial interactions that may or may not involve fitness costs.

### Predicting ecological patterning in simple communities using a generalized fitness model

To search for ecological patterning rules in microbial communities, we built a three-dimensional fitness model that ignored molecular details and instead focused on the fitness effects of interactions ('The fitness model' in 'Materials and methods'). Specifically, two populations of cells, marked as red and green, were initially randomly distributed on a surface (*Figure 1A*). Cells grew horizontally until sufficiently confined, at which point they grew upward to accommodate the birth of new cells (*Figure 1—figure supplement 1*, *Video 1*). Thus, no active cell motility was present during community growth. The growth rate of a focal cell was determined by its basal fitness as a single cell and by its interactions with other cells in a defined interaction neighborhood. To reflect the negative fitness effects of intra- and inter-population competition for shared resources ([~ ~]), the fitness of the focal cell was decreased as the total population size in the interaction neighborhood increased. In addition, the focal and partner cells affected each other's fitness positively (↑), negatively (↓), or neutrally (~). The magnitude of fitness effect is quantitative. Thus, to obtain qualitative ecological patterning rules, we focused on strong interactions in which ↑ and ↓ exert fitness effects large enough to be comparable to the recipient's basal fitness. In most simulations using the fitness model, the basal fitnesses of both partners were non-zero, and therefore ↑ represented strong facultative interactions: a participant could survive on its own at its basal fitness but fared much better in the presence of its partner.

We first analyzed the population composition of communities formed in the fitness model. Simulations (*Figure 1—figure supplement 2* and *Figure 1—source data 1*) and analytical calculations

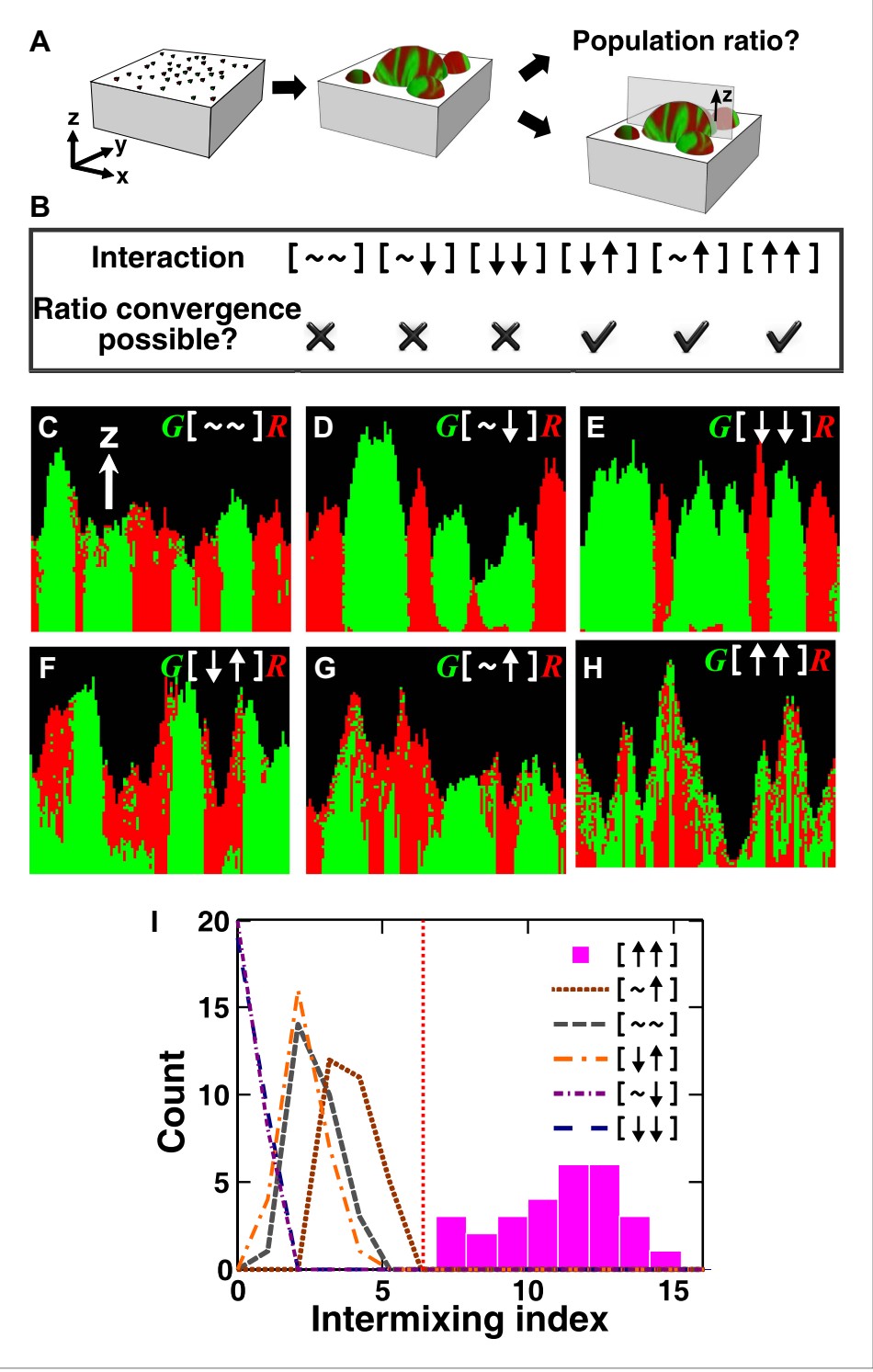

**Figure 1**. The fitness model generates two ecological patterning predictions. (**A**) In all simulated and experimental communities (see 'Materials and methods'), two populations of cells, marked in red and green, were initially randomly distributed on a surface unless otherwise stated. The two populations engaged in one of the six ecological interactions. Population ratios for the entire community and patterns of vertical cross sections were examined. (**B**) The fitness model predicts that strong interactions beneficial to at least one partner can potentially lead to the convergence of initially disparate population ratios (**Figure 1—figure supplement 2**). (**C**)–(**H**) Representative vertical cross-sections of simulated communities, each engaging in one of the six types of

*Figure 1. Continued on next page*

*Figure 1. Continued*

ecological interactions, are presented. The fitness effects of ↑ and ↓ are large compared to the non-zero basal fitness of the recipient (**Figure 1—source data 1**), and therefore [↑ ↑] is strong facultative cooperation.

(**I**) Vertical cross-sections of single-cell thickness from cooperative communities show significantly more intermixing than those from other communities (*n* = 28 sections; p<0.01, Mann–Whitney *U* test). An intermixing index of 6 (red dotted line) or above separates cooperative from non-cooperative communities in our simulations. To reduce the correlation of sections sampled from the same community, nearest sections were separated by at least seven sections.

The following source data and figure supplements are available for figure 1:

**Source data 1.** Parameter values used in the fitness model.

**Figure supplement 1**. Cell rearrangement in simulations follows experimental observations on cells that are not actively motile.

**Figure supplement 2**. The fitness model predicts that convergence of population ratios is possible when an interaction benefits at least one partner.

**Figure supplement 3**. Strong mutual antagonism can lead to rapid divergence of population ratios.

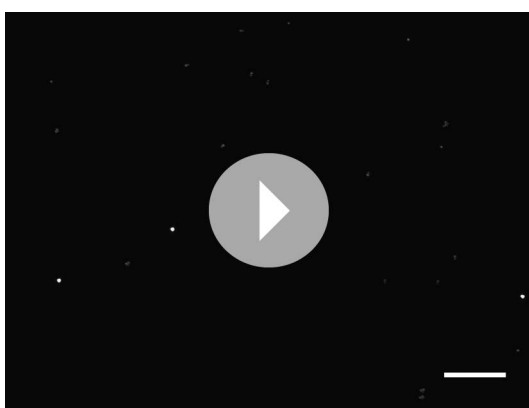

**Video 1**. Yeast cells bud to the sides when there is available space and bud upward when sufficiently confined (corresponding to **Figure 1—figure supplement 1A**). To infer the process of cell rearrangement in three-dimensional communities, we monitored how single YFP-fluorescent yeast cells grew into microcolonies on top of solid agarose. Initially, dividing cells pushed other cells to the side such that all cells remained in the same plane. When a cell was sufficiently confined from the sides by other cells (approximately within a 5-cell radius), it could no longer bud to the side and instead budded upward, as indicated by higher intensities in the fluorescence images. Continued growth of microcolony forced more cells in the middle to send their progeny to upper layers, while cells close to the edge could still push other cells to the side and remain on the agarose surface. All images are taken with the same exposure time. Scale bar is 20 μm.

('Requirements for steady-state ratios in the six types of communities' in 'Materials and methods') show that for interactions benefiting at least one partner, including cooperation ([↑ ↑]), commensalism ([~ ↑]), and exploitation ([↓ ↑]), different initial partner ratios can potentially converge over time (**Figure 1B**). Ratio convergence requires 'balanced' fitness. For example, for A[~ ↑]B to achieve ratio convergence, the basal fitness of A must be higher than that of B and after gaining the fitness benefit from commensalism, B must be able to grow at least as fast as A. This way, a situation incompatible to ratio convergence, that is, one partner always fitter than the other, does not occur. Using engineered competitive, obligatory commensal, and obligatory cooperative yeast communities (see below), we confirmed that population ratios converged in the commensal and cooperative but not competitive communities (**Figure 2—figure supplement 1**). This convergence of population ratios reflects a balance between supply and consumption when growth is limited by supply (**Shou et al., 2007**): if, for instance, the supplier population suddenly increased in relative abundance, then each individual in the consumer population would receive more benefit and grow faster, which would return the ratio to its original value.

We next examined patterns in vertical cross-sections of communities simulated by the fitness model (**Figure 1A**, parameters in **Figure 1—source data 1**), because patterning along the *x* and *y* directions can depend on the initial spatial distribution of cells whereas patterning along the vertical *z* direction results from growth under the fitness influences of ecological interactions. In [~ ~], [~ ↓], and [↓ ↓], red and green populations primarily formed columns that are spatially segregated from each other (**Figure 1C–E**). In [↑ ↓] and [~ ↑], frequently one of the populations (green) either

formed a column or became covered by the partner population (red) (*Figure 1F–G*). Only in cooperation ([↑ ↑]) conferring large fitness benefits to both partners, the two partner populations appeared to be extensively 'intermixed', manifested as the two different cell types successively piling on top of each other (*Figure 1H*).

To compare levels of intermixing in different communities, we defined an 'intermixing index' as the average number of cell type transitions spanning community height ('Spatial analysis' in 'Materials and methods'). Since the intermixing index can be a function of community height, we compared intermixing indexes of simulated communities at equivalent heights. Statistically significant differences were observed between strong cooperation versus other types of interactions (*Figure 1I*). Thus, we predicted that partner intermixing would distinguish strong cooperation from other ecological interactions.

## Partner intermixing in engineered obligatory cooperative yeast communities and a corresponding diffusion model

To test the prediction that strong cooperation is the only ecological interaction capable of driving partner intermixing, we engineered yeast communities engaged in competitive ([~ ~]), obligatory commensal ([~ ↑]), and obligatory cooperative ([↑ ↑]) metabolic interactions. Competitive communities represented the baseline intra- and inter-population competition common to 'all' communities, while commensalism served as the most stringent control to be discriminated against.

All engineered yeast communities consisted of two non-mating *S. cerevisiae* strains, a *G* strain expressing GFP or YFP and an *R* strain expressing DsRed (see 'Materials and methods'). In competitive communities, prototrophic *R* and *G* competed for shared nutrients in agarose and for space (*Figure 2—figure supplement 1A*). Depending on their genetic backgrounds, the two strains engaged in either equal-fitness or unequal-fitness competition. In obligatory commensal communities, $R_{\to A}^{\leftarrow L}$ took in lysine from the media and overproduced adenine to feed the adenine-requiring $G^{\leftarrow A}$ (*Figure 2—figure supplement 1B*). In obligatory cooperative communities (previously described as Cooperation that is Synthetic and Mutually Obligatory, or "CoSMO"), $R_{\to A}^{\leftarrow L}$ overproduced adenine and required lysine while $G_{\to L}^{\leftarrow A}$ overproduced lysine and required adenine (*Shou et al., 2007*) (*Figure 2—figure supplement 1C*). When mixed, the two cooperative strains could form a viable community, growing from low to high densities in synthetic minimal medium (SD) lacking adenine and lysine (*Shou et al., 2007*).

To predict and extrapolate experimental results of yeast communities, we developed a three-dimensional model based on the consumption, release, and diffusion of metabolites in the yeast communities ('the diffusion model' in 'Materials and methods'). Specifically, the diffusion model assumed that metabolites diffused in the community and agarose (*Figure 2—figure supplement 2*) and that cell growth depended on the local concentration of its limiting metabolite according to Monod's equation (*Monod, 1949*). Most parameters in the diffusion model were measured experimentally (*Figure 2—source data 1*). We reasoned that if predictions from the diffusion model were consistent with experimental observations, we could use this model to simulate experiments that would be technically difficult to perform.

To examine vertical patterning in yeast communities, we first used top-view time-lapse images to infer patterns which were subsequently verified by cryosectioning. This is because confocal and two-photon microscopy cannot penetrate deep into yeast communities (*Váchová et al., 2009*). In competitive communities, whether in the diffusion model or experiments, time-lapse top-views suggested population segregation. For equal-fitness competitive communities (*Figure 2A*, left; *Video 2*), individual cells (i) grew into microcolonies (ii) which continued to grow and expand until they reached neighboring microcolonies (iii). After this stage, cells were unable to push other cells to the side, and further cell divisions mainly occurred in the vertical *z* direction (*Figure 1—figure supplement 1*). Consequently, columns of primarily a single cell type formed, and top-views of the community remained static (compare iv and v). Competitive communities composed of populations with different fitnesses developed similarly (*Figure 2A*, right; *Video 3*), except that the fitter population expanded in the top view during growth (compare the green population in iv and v).

In obligatory commensal communities, time-lapse top-views suggested a 'layered' pattern with one population covering the other (*Figure 2B*; *Video 4*). We found that in both the diffusion model and experiments, $R_{\to A}^{\leftarrow L}$ initially grew rapidly by consuming lysine in the agarose (compare i, ii, and iii). In contrast, $G^{\leftarrow A}$ initially grew slowly, presumably limited by the low level of adenine released by $R_{\to A}^{\leftarrow L}$

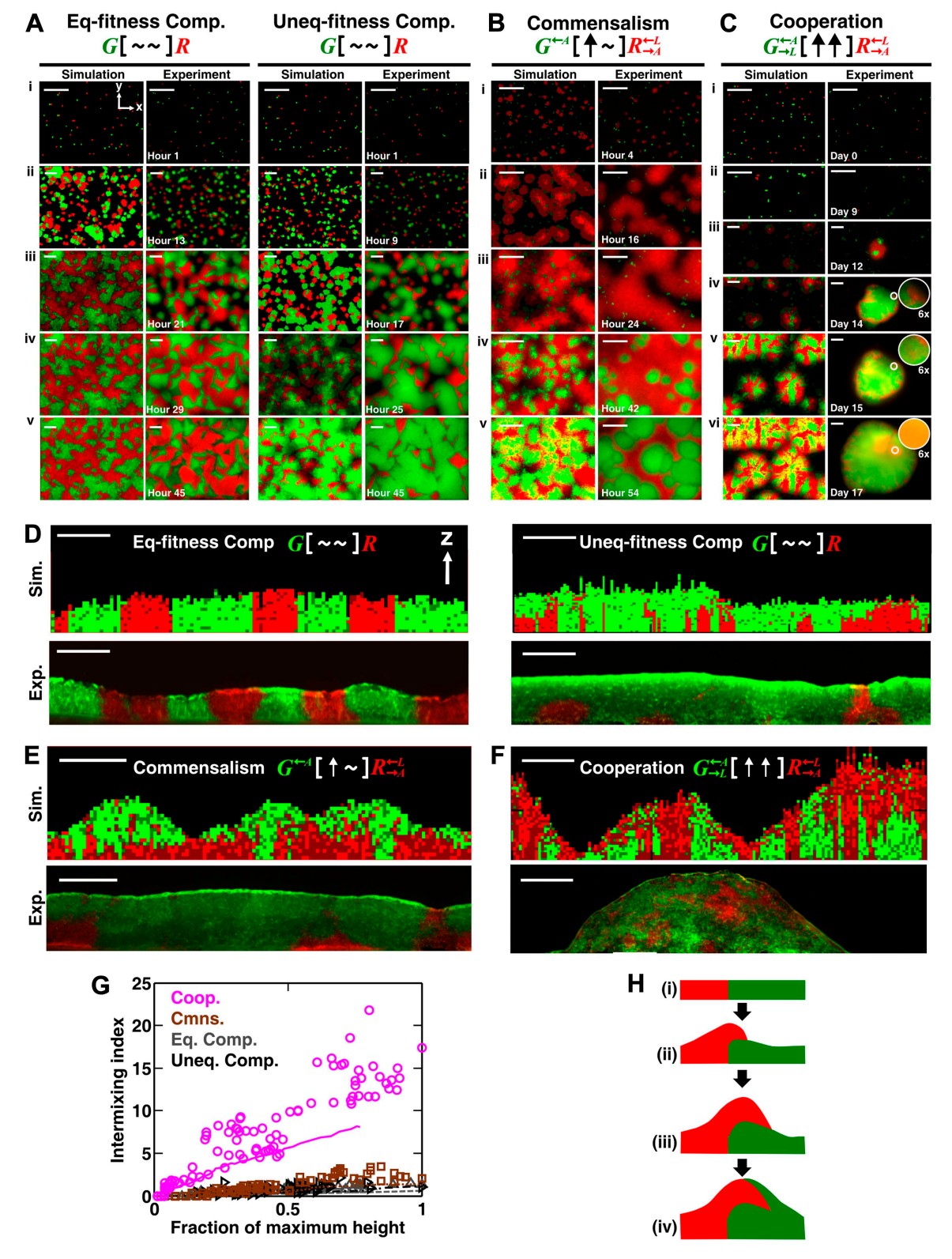

**Figure 2**. Obligatory cooperation, but not competition or obligatory commensalism, results in substantial partner intermixing in engineered yeast communities and in communities simulated using the diffusion model. Competitive communities of strains with equal fitness (equal-fitness competition, abbreviated as 'Eq-fitness Comp.') showed population segregation as suggested by static late-stage top-views (**A**, left) and columnar cross-section

*Figure 2. Continued on next page*

*Figure 2. Continued*

patterns (**D**, left). When competing strains had different fitnesses (unequal-fitness competition, abbreviated as 'Uneq-fitness Comp.'), the fitter population *G* expanded during growth, as evident in top views (**A**, right) and vertical cross-sections (**D**, right). In obligatory commensal communities, since one supplier could support the birth of multiple consumers, consumers eventually overgrew and covered suppliers (top-views in **B** and vertical cross-sections in **E**). Obligatory cooperative communities showed substantial population intermixing as suggested by alternating cell types in top-views (**C**, 6× magnification insets in experiments) and patchy cross-section patterns (**F**). Top views of communities from the diffusion model integrate intensity and color over height such that brighter colors represent higher cell numbers and yellowness indicates the simultaneous presence of green and red. Scale bar: 100 μm. All communities started from total 500 cells/mm² and *R*:*G* = 1:1. (**G**) Quantification of intermixing in experimental (symbols) and diffusion-model (lines) communities showed that while the intermixing index remained low for commensal (brown) and competitive (grey and black) communities, it increased linearly with community height in obligatory cooperative (magenta) communities. Results from the diffusion model underestimated intermixing indexes because a confined cell was modeled to divide strictly vertically upward (***Figure 1—figure supplement 1***) without allowing cell movements in horizontal directions (***Figure 3—figure supplement 1F***). (**H**) A conceptual model illustrates the development of intermixing over time in a strongly cooperative community with 1:1 steady-state population ratio. Local deviations from the steady-state ratio result from asymmetric partner properties and/or stochastic fluctuations in cell numbers (i). The under-abundant population (red) grows faster than its neighboring over-abundant partner (green). Due to the spatial localization of large cooperative benefits, cells near population borders grow faster than those farther away. Consequently, cells from the initially under-abundant red population at the border divide the fastest. Progeny that pile on the green partner have more access to cooperative benefits than those who do not (ii), which favors intermixing. When the previously over-abundant partner becomes under-abundant (iii), piling-up in the opposite direction occurs (iv, green on red).

The following source data and figure supplements are available for figure 2:

**Source data 1.** Definitions and values of parameters used in the diffusion model.

**Figure supplement 1**. In engineered yeast communities, obligatory cooperation and obligatory commensalism allow initially different partner ratios to converge over time.

**Figure supplement 2**. Basic assumptions in the diffusion model.

**Figure supplement 3**. Cooperative communities exhibit a characteristic patch size associated with the spatial localization of benefits.

**Figure supplement 4**. Obligatory cooperative yeast partners intermix.

before $R_{\to A}^{\leftarrow L}$ became abundant (compare i, ii, and iii). During the growth of $R_{\to A'}^{\leftarrow L}$ some $G^{\leftarrow A}$ cells had been pushed to the top layer of the community (iii). As the $R_{\to A}^{\leftarrow L}$ population continued to expand and release adenine, $G^{\leftarrow A}$ started to grow rapidly (compare iii and iv). Eventually, $R_{\to A}^{\leftarrow L}$ stopped growing after lysine in the agarose had been depleted. Since the amount of adenine released during the lifetime of every $R_{\to A}^{\leftarrow L}$ cell could support the birth of multiple $G^{\leftarrow A}$ cells (***Shou et al., 2007***), $G^{\leftarrow A}$ population outnumbered and covered $R_{\to A}^{\leftarrow L}$ (v).

In contrast, time-lapse top-views of obligatory cooperative communities suggested population intermixing (***Figure 2C***; ***Video 5***). After plating on agarose lacking adenine and lysine (i), cells in both populations divided once or twice by utilizing metabolites stored in their vacuoles (***Shou et al., 2007***). $R_{\to A}^{\leftarrow L}$ released adenine and entered the death phase while $G_{\to L}^{\leftarrow A}$ continued to grow by utilizing the released adenine (ii) (***Shou et al., 2007***). $G_{\to L}^{\leftarrow A}$ entered the death phase and released lysine after a significant delay due to their better starvation tolerance compared to $R_{\to A}^{\leftarrow L}$ (***Shou et al., 2007***). Released lysine supported the growth of surviving $R_{\to A}^{\leftarrow L}$ cells into microcolonies (iii). $R_{\to A}^{\leftarrow L}$ in turn released adenine which promoted growth of nearby $G_{\to L}^{\leftarrow A}$ cells and led to partial covering up of $R_{\to A}^{\leftarrow L}$ microcolonies by rapidly growing $G_{\to L}^{\leftarrow A}$ (iv and v, insets). Abundant $G_{\to L}^{\leftarrow A}$ cells provided enough lysine for rapid growth of local $R_{\to A}^{\leftarrow L}$ cells, which gave rise to patches of $R_{\to A}^{\leftarrow L}$ cells on top of the community (vi, inset). These community growth kinetics were consistent with previous measurements in liquid cultures (***Shou et al., 2007***).

To confirm that partners intermixed in cooperative but not non-cooperative communities, we obtained vertical cross-sections of competitive, commensal, and cooperative communities at their maximal community heights ('Cryosectioning' for experimental communities in 'Materials and methods'). We found that in both the diffusion model and experiments, equal-fitness competitive communities formed segregated columns (***Figure 2D***, left). Unequal-fitness (***Figure 2D***, right) and obligatory commensal communities (***Figure 2E***) displayed a layered pattern in which the top

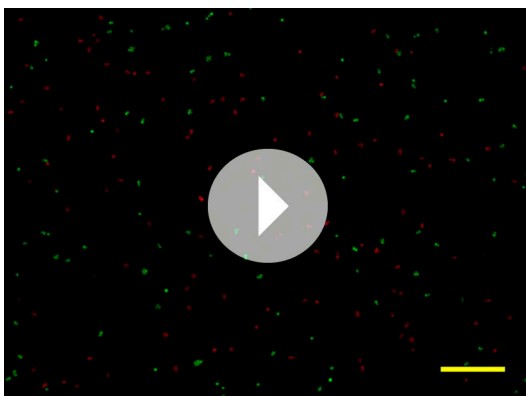

**Video 2**. Top views of an equal-fitness competitive community suggest population segregation (corresponding to *Figure 2A*, left). Competitive communities of strains with equal fitness showed population segregation as suggested by static late-stage top-views. The community started from a uniform distribution of total 500 cells/mm² and R:G = 1:1. Intensities of both fluorescent channels in images at different times were normalized to the same maximum value for better representation of patterns throughout growth. Blanks in the video were due to removal of the dish to sample other replicate communities for flow-cytometry or sectioning. Scale bar is 100 µm.

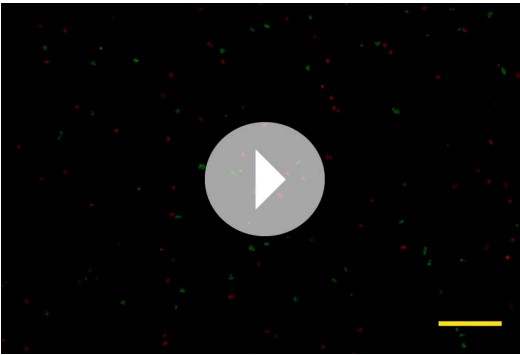

**Video 3**. In unequal-fitness competition, the fitter population gradually covers the less fit population (corresponding to *Figure 2A*, right). Here, G is fitter than R. The community started from a uniform distribution of total 500 cells/mm² and R:G = 1:1. Intensities of both fluorescent channels in images at different times were normalized to the same maximum value for better representation of patterns throughout growth. Scale bar is 100 µm.

portion of a community was dominated by one partner. In contrast to competitive and obligatory commensal communities, obligatory cooperative communities appeared to show population intermixing with patches of red and green cells emerging on top of each other (*Figure 2F* and *Figure 2—figure supplement 4A and B*). Cooperative intermixing appeared to be less in the diffusion model than in yeast communities. This is presumably because the diffusion model assumed that a confined cell would bud strictly upward whereas in yeast communities, cell divisions could stray to the side (*Figure 3—figure supplement 1F*).

To compare levels of intermixing in different communities, we quantified the intermixing index of each community (*Figure 2G*, symbols for experiments and lines for results from the diffusion model). As expected, equal-fitness competition resulted in small intermixing indexes at all community heights (*Figure 2G*, grey). In unequal-fitness competition (*Figure 2G*, black) and obligatory commensalism (*Figure 2G*, brown), the formation of a layered pattern caused a small increase in the intermixing index that subsequently leveled off. In contrast, the intermixing index in obligatory cooperative communities increased proportionally to community height in both experiments and the diffusion model (*Figure 2G*, magenta). This proportionality suggested the existence of a characteristic patch size ('Spatial analysis' in 'Materials and methods'), denoted $\lambda_z^*$, of 10–20 µm. The characteristic patch size was independent of initial conditions (*Figure 2—figure supplement 3A*). Indeed, a calculation of the patch size based on experimentally-determined release, diffusion, and consumption of exchanged metabolites yielded comparable results ('Calculating the characteristic patch size in cooperative yeast communities' in 'Materials and methods').

What caused partner intermixing during cooperation? Using the diffusion model, we found that if cooperative benefits were not spatially localized because of instant distribution of benefits throughout the community or because of excessive supply levels, intermixing was diminished (*Figure 2—figure supplement 3B*). Based on this result and based on patterns observed in top-views and vertical cross-sections of cooperative communities, we propose that local deviations from the steady-state ratio coupled with localized large cooperative benefits cause partners to 'take turns' to grow, which leads to population intermixing (*Figure 2H*). In summary, in communities engaging in strong cooperation, but not in communities governed by other types of ecological interactions, the intermixing index increases proportionally as a function of community height. This proportionality is due to a fixed patch size determined by localized nutrient supply and consumption.

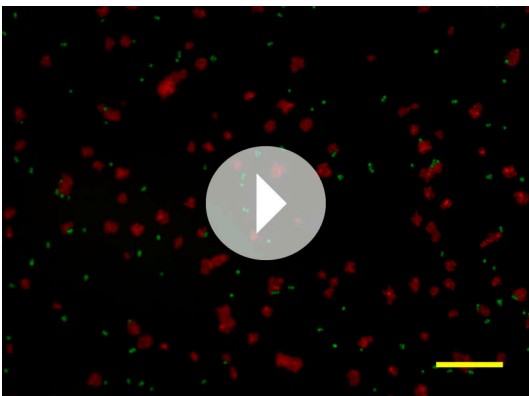

**Video 4**. Top views of an obligatory commensal community suggest population layering (corresponding to *Figure 2B*). For a detailed explanation of the growth kinetics of the community ($R_{\to A}^{\leftarrow L}$ [~↑] $G^{\leftarrow A}$), please refer to the main text. The community started from a uniform distribution of total 500 cells/mm$^2$ and $R_{\to A}^{\leftarrow L}$:$G^{\leftarrow A}$ = 1:1. Intensities of both fluorescent channels in images at different times were normalized to the same maximum value for better representation of patterns throughout growth. Blanks in the video were due to removal of the dish to sample other replicate communities for flow-cytometry or sectioning. Scale bar is 100 μm.

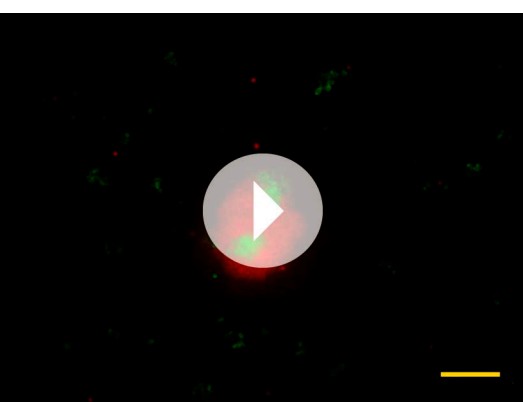

**Video 5**. Top views of an obligatory cooperative community suggest populations intermixing (corresponding to *Figure 2C*). For a detailed explanation of the growth kinetics of the community ($R_{\to A}^{\leftarrow L}$ [↑ ↑] $G_{\to L}^{\leftarrow A}$), please refer to the main text. The community started from a uniform distribution of total 500 cells/mm$^2$ and $R_{\to A}^{\leftarrow L}$:$G_{\to L}^{\leftarrow A}$ = 1:1. Intensities of both fluorescent channels in images at different times were normalized to the same maximum value for better representation of patterns throughout growth. Blanks in the video were due to removal of the dish to sample other replicate communities for flow-cytometry or sectioning. The video started at ~300 hr, after the formation of $R_{\to A}^{\leftarrow L}$ microcolonies. Scale bar is 100 μm.

## Strongly cooperative partners intermix under a wide range of conditions

Under what conditions can we observe partner intermixing in cooperation? First, we experimentally tested partner intermixing in obligatory yeast cooperative communities initiated under different population ratios and densities. Next, we used the diffusion model to examine communities in which obligatory cooperative partners interacted with different dynamics. Finally, using the fitness model, we tested intermixing in facultative cooperation by varying the relative magnitude of cooperative benefits compared to the basal fitness of the two interacting populations. We utilized yeast communities when experimentally possible and otherwise took advantage of the diffusion and the fitness models.

We found that intermixing was insensitive to initial conditions in the yeast obligatory cooperative communities. The initial partner ratio did not significantly affect the level of intermixing (*Figure 3—figure supplement 1D*). At very high initial cell densities, we observed significant intermixing in communities directly above the inoculation area even in the absence of cooperation (*Figure 3—figure supplement 1E*, yellow shading). This is because high cell densities put different cell types in close proximity to one another, and cell divisions that were not perfectly straight upward (*Figure 3—figure supplement 1F*) caused intermixing. However, we reasoned that communities beyond the inoculation area might reveal patterns indicative of the underlying interactions, because these regions are formed by cell growth under the fitness influences of interactions. To test this possibility, we spotted cell mixtures at high densities on agarose and allowed the community to expand to new territories beyond the inoculation area. Even though communities directly above the inoculum always appeared highly intermixed (*Figure 3A*, 'Center' and *Figure 2—figure supplement 4C*), in edge sections, significant intermixing was only observed in cooperative communities (*Figure 3A*, 'Edge' and *Figure 2—figure supplement 4D*).

Intermixing is insensitive to interaction dynamics, so long as large cooperative benefits remain sufficiently localized for both partners. As described above, in the diffusion model, excessive supply amounts or instant distribution of benefits throughout the community diminished intermixing (*Figure 2—figure supplement 3B*). The former can occur if the local availability of cooperative benefits is not growth-limiting. However, cooperative benefits are unlikely to be available in large excess because of the potential fitness cost of generating benefits and because of competition

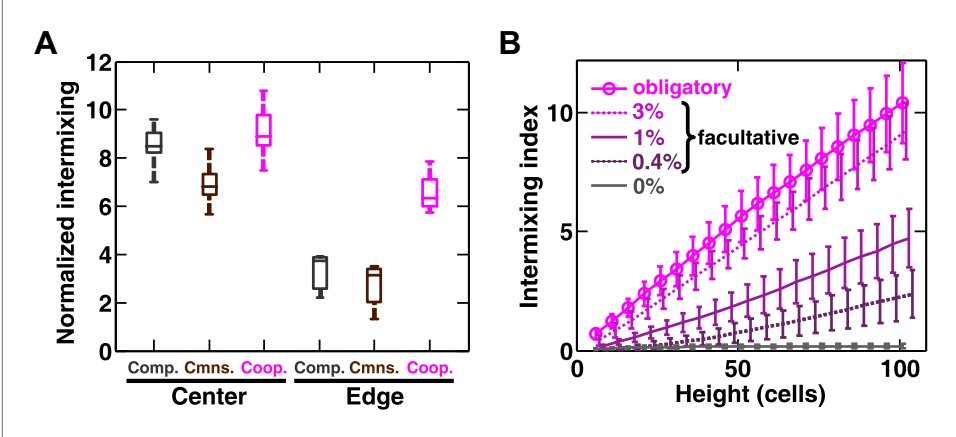

**Figure 3**. Strongly cooperating populations intermix under a wide range of conditions. (**A**) In engineered yeast communities, even though both obligatory cooperative and non-cooperative communities directly above the high-density inoculation spot showed high population intermixing ('Center'), edge sections ('Edge') of obligatory cooperative communities showed significantly more intermixing than those from non-cooperative communities (Mann–Whitney $U$ test, $p < 5 \times 10^{-5}$). A total of $10^6$ $R$ and $G$ cells at a 1:1 ratio were deposited in an inoculation spot of ~2 mm$^2$, corresponding to 10 cell layers. Communities were allowed to grow and expand beyond the inoculation spot on a 4-mm-tall agarose pad of 500 mm$^2$ area. Vertical sections from the edges were taken at a height approximately half of the maximum community height at the center. Box plots show the 25th to 75th percentile range, with the median marked with a line and whiskers extending to the 95% confidence interval. (**B**) In simulations using the fitness model, facultative cooperation conferring smaller fitness benefits required greater community heights to exhibit a significant level of intermixing. The strength of facultative cooperation is shown as the ratio of fitness benefit received from each cooperative partner cell in the interaction neighborhood to the basal fitness of the focal cell. Simulation parameters can be found in *Figure 3—source data 1*. Error bars indicate 95% confidence interval.
The following source data and figure supplements are available for figure 3:

**Source data 1.** Parameter values used in the fitness model in *Figure 3B*.
**Figure supplement 1.** Intermixing is observed in obligatory cooperative communities over a wide range of conditions.

for these benefits in the partner population. As far as diffusion is considered, the diffusion model showed significant population intermixing in obligatory cooperative communities with diffusion constants varying more than 10-fold (*Figure 3—figure supplement 1A*). Additional calculations showed that intermixing was largely insensitive to diffusion kinetics, because the characteristic patch size was related to the diffusion constant by 1/5 power ('Calculating the characteristic patch size in cooperative yeast communities' in 'Materials and methods'). The diffusion model also showed that intermixing was present with or without a delay in $G_{\rightarrow L}^{\leftarrow A}$ supplying lysine benefits to the partner (*Figure 3—figure supplement 1B*). Furthermore, even though asymmetry between properties of partners (*Figure 2C*) would seem to facilitate intermixing (*Figure 2H*), asymmetry in partner properties was not required to generate intermixed patterns: partners that grew, died, and released and consumed metabolites at identical rates intermixed (*Figure 3—figure supplement 1C*).

Using the fitness model, we found that in facultative cooperation, small fitness benefits generated less intermixing than large fitness benefits (*Figure 3B*). This result is intuitive: facultative cooperation should resemble obligatory cooperation if fitness benefits are large for both partners. When only one partner receives a large fitness benefit, facultative cooperation should resemble obligatory commensalism. Finally, small fitness benefits for both partners will make facultative cooperation resemble competition. In facultative cooperation with smaller fitness benefits, intermixing would be apparent if communities could grow to greater heights. Further experiments are required to test these predictions. Together, these results suggest that intermixing relies on spatial localization of cooperative benefits that are sufficiently large to both partners, and is otherwise insensitive to initial conditions or the detailed kinetics of interactions.

## Communities of naturally mutualistic microbes exhibit intermixing

To test whether cooperative patterning applies to other biological systems, we examined redox-coupling in a two-species methane-producing biofilm consisting of the bacterium *Desulfovibrio vulgaris* and the archaeon *Methanococcus maripaludis*. In the absence of sulfate and hydrogen, the two species engage in obligatory mutualism: *D. vulgaris* ferments lactate and promotes the growth of *M. maripaludis* by supplying the electron donor $H_2$, while *M. maripaludis* promotes the growth of *D. vulgaris* by consuming the $H_2$ gas which can be inhibitory to *D. vulgaris* under these conditions (*Figure 4A*). Similar types of syntrophic interactions leading to methane production typically occur in microbial consortia that digest organic compounds in freshwater sediments, sewage treatment plants, and the guts of ruminants (*Conrad et al., 1985*; *Schink, 1997*).

Vertical cross-sections of independent *D. vulgaris*–*M. maripaludis* biofilms indeed exhibited increasing intermixing as a function of community height (*Figure 4B*). Thus, naturally mutualistic microbes cooperating through a coupling mechanism different from metabolite exchange also exhibited a significant level of intermixing. Other known cooperative communities, including those degrading herbicide pollutants (*Breugelmans et al., 2008*) and those colonizing teeth (*Palmer et al., 2001*) also seemed to display intermixed patterns, although we do not know whether the intermixing indexes of these communities increased linearly with height.

## Most of the strongly-cooperative pairs intermix in simulated multi-species communities

In communities with more than two species, indirect interactions can obscure direct interactions (*Wootton, 2002*). For example, if A promotes B which inhibits C, it will appear that A inhibits C. This is akin to indirect interactions between gene products in a cell. To test whether intermixing between cooperators was affected by other members of a community, we used the fitness model to simulate communities composed of five interacting species (*Figure 5* and *Figure 5—figure supplement 1*). We randomly assigned one of the six possible ecological interactions between each pair of populations, and consequently, each network had 10 pairwise interactions. The fitness effects of ↑ and ↓ were sufficiently large to be comparable to the recipient's basal fitness. A total of 240 pair-wise interactions in 24 independent communities were examined in the fitness model. In most cases (26 out of 31), cooperative pairs intermixed (*Figure 5A,D*). Occasionally, commensal pairs (*Figure 5B*, ② [~ ↑] ③) showed substantial intermixing (3 out of 58 commensal pairs, *Figure 5D*), or cooperative pairs (e.g., *Figure 5C*, ① [↑ ↑] ③) showed little intermixing (5 out of 31, *Figure 5D*), presumably due to interactions through other community members (*Figure 5B*, ③ indirectly promoted ② through ④; *Figure 5C*, ③ promoted ⑤ which inhibited ①). These deviations are consistent with the notion that the intermixing index reflects the 'overall' interaction between two partners, integrating any additional fitness effects of indirect interactions through other community members. Thus, strongly cooperating partners intermix while deviations from this expectation reflect the presence of strong indirect interactions.

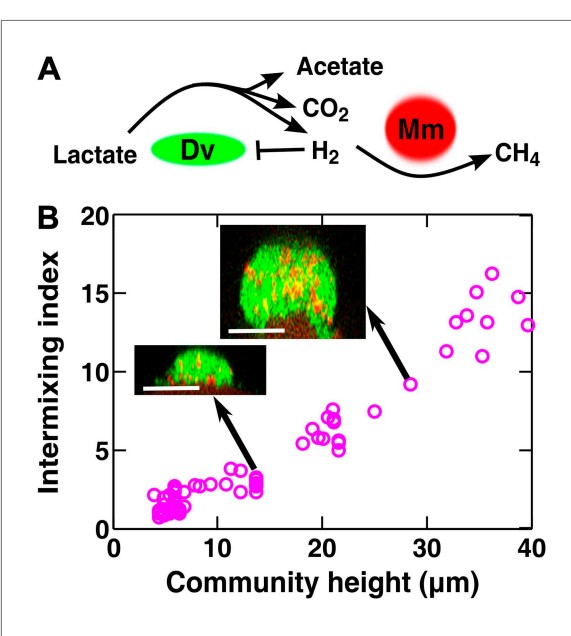

**Figure 4**. Obligatory cooperation through redox-coupling leads to partner intermixing. (**A**) In the absence of sulfate and hydrogen, the bacterium *Desulfovibrio vulgaris* (Dv) and the archaeon *Methanococcus maripaludis* (Mm) cooperate through redox coupling. Dv ferments lactate and produces mainly acetate, $CO_2$, and $H_2$. However, this reaction is not thermodynamically favorable unless $H_2$ is kept at very low concentrations. $H_2$ is used by Mm to reduce $CO_2$ to methane. (**B**) In cooperative biofilms of Dv (green) and Mm (red), the intermixing index increased with height. Cell identification relied on FISH (see 'Materials and methods'). Scale bar: 20 µm.

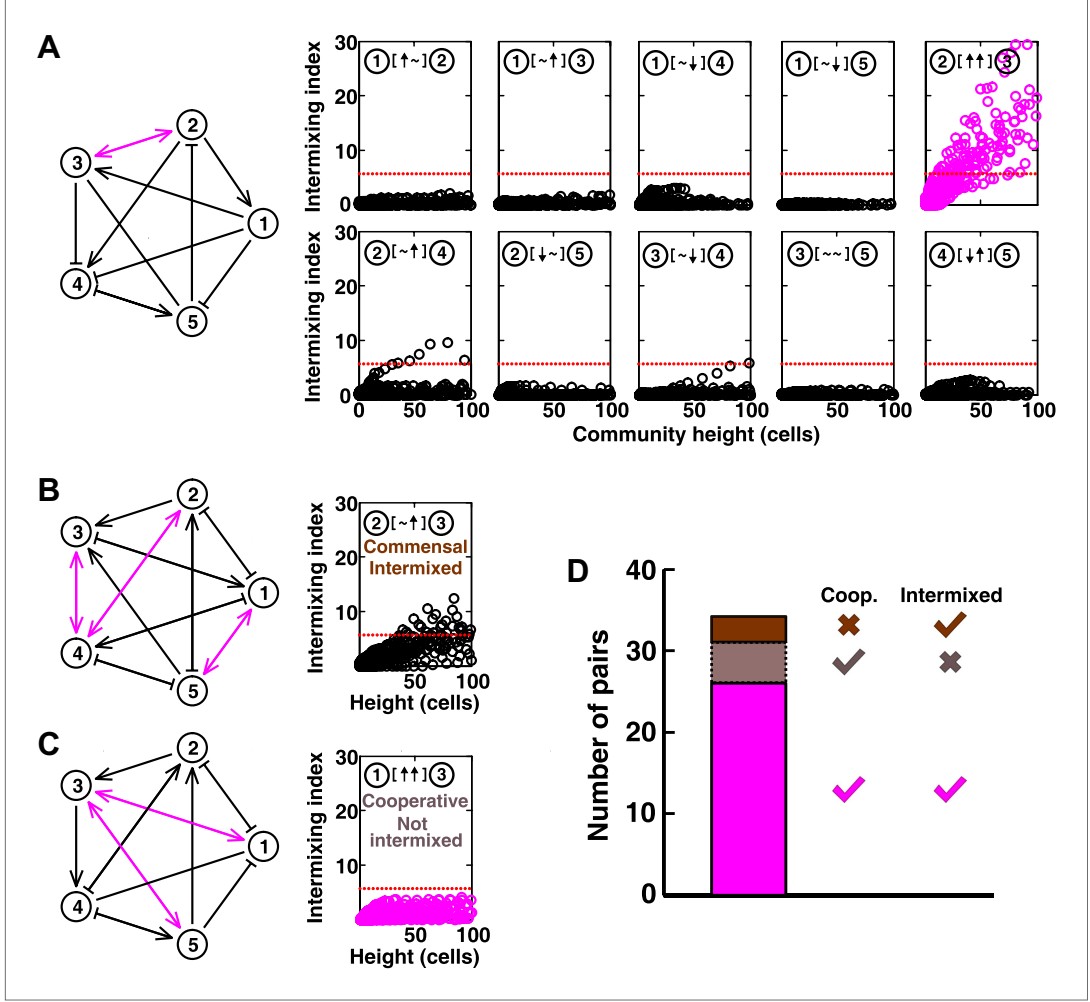

**Figure 5**. Most of the strongly-cooperative pairs intermixed in simulated five-species communities. (**A-C**) Examples of networks in which cooperative pairs intermixed (**A**), non-cooperative pairs intermixed (**B**), or cooperative pairs did not intermix (**C**) are shown. In the schematic network diagrams, line termini of →, ⊣, and — represent ↑, ↓, and ~, respectively; cooperative pairs are highlighted in magenta. Simulations were performed using the fitness model. The basal fitness for each population was chosen randomly from a range spanning 0.03–0.05/hr. The fitness effect from each partner cell in the interaction neighborhood was either 0 for ~, or otherwise randomly chosen to be approximately 2–3% of basal fitness to achieve strong interaction. Initial population ratios were randomly assigned such that no population was initially lower than 5% of the total population. Simulations were run for 10 generations, and vertical cross-sections of the final communities were examined for intermixing. We considered intermixing index exceeding a threshold of 6 (**Figure 1I**) as significant (red dotted lines). The remaining nine panels in **Figure 5B and C** are provided in **Figure 5–figure supplement 1**. (**D**) Quantifying intermixing in a total of 240 interactions from 24 independent communities showed that most of the cooperative pairs intermixed (magenta). Indirect interactions through other community members could make cooperative pairs not intermixed (grey) or non-cooperative (commensal) pairs intermixed (brown).

The following figure supplements are available for figure 5:

**Figure supplement 1**. The complete results of **Figure 5B and C** (panel A and B, respectively).

# Discussion

## Using controlled systems to identify patterns driven by ecological interactions

Patterning is driven by cell–cell and cell–environment interactions. Here, we examined how the net fitness effects of cell–cell interactions could influence patterning. Using defined mathematical and

biological systems under controlled spatial environments, we have established two expectations for ecological patterning between two interacting partners.

The first expectation is that interactions conferring large fitness benefits to at least one partner ([↑ ↑], [~ ↑], and [↓ ↑]) can potentially result in ratio convergence (*Figure 1—figure supplement 2* and *Figure 2—figure supplement 1*). Ratio convergence confers stability to community composition. The composition of a community is defined to be locally (or globally) stable if after small (or any-size) deviations from it, either due to initial conditions different from it or perturbations, the community eventually returns to this composition. An analytical calculation based on the fitness model showed that in [↑ ↑] and [~ ↑], global stability can be achieved whereas in [↓ ↑], local stability can be achieved (*Figure 1—figure supplement 2*; 'Requirements for steady-state ratios in the six types of communities' in 'Materials and methods'). Experimentally, obligatory cooperation and obligatory commensalism led disparate initial ratios to converge (*Figure 2—figure supplement 1*). In contrast, in competition between two populations of equal fitness, population ratio is determined by the initial value until a perturbation resets the value which will remain unchanged until the next perturbation. In competition between two populations of unequal fitness, the fitter population should increase in frequency and therefore, population ratio is unstable. In amensalism and mutual antagonism, population ratios at best have an unstable fixed point (*Figure 1—figure supplement 2B*). This means that even if the population ratio may seem 'fixed', slight deviations will send it farther and farther away from the original fixed value. The reason for this instability, using mutual antagonism as an example, is because if A inhibits B and B inhibits A (A[↓ ↓]B), then an increase in population A will facilitate inhibition of B and therefore make B less able to inhibit A, further increasing the relative abundance of A.

The second expectation is that strong cooperation results in partner intermixing in the direction vertical to the surface of initial colonization. Cooperative intermixing requires spatially localized benefits that are sufficiently large to both partners (*Figure 2—figure supplement 3* and *Figure 3B*), and otherwise appears to be robust against variations in initial conditions (*Figure 2—figure supplement 3A*, *Figure 3A*, and *Figure 3—figure supplement 1D*) or reaction dynamics (*Figure 3—figure supplement 1A–C*). We have observed cooperative intermixing in the fitness model simulating strong facultative cooperation (*Figure 1*) and in obligatory experimental systems including engineered yeast communities (*Figure 2*) and syntrophic biofilms (*Figure 4*). The intermixing indexes in these communities increased linearly as a function of community height, because these communities exhibited fixed-size patches associated with the spatial localization of benefits (*Figure 2—figure supplement 3*; 'Calculating the characteristic patch size in cooperative yeast communities' in 'Materials and methods'). This is in stark contrast with other ecological interactions which lead to segregated or layered patterns (*Figures 1 and 2*) in which the intermixing index remained low or increased transiently before leveling off. In simulated multi-species communities, strongly cooperative pairs intermixed in most cases; cases where cooperative pairs did not intermix or non-cooperative pairs intermixed were likely caused by strong indirect interactions through other partners. In a theoretical study using a one-dimensional stepping-stone model, strongly cooperating partners also appeared much more mixed than competing partners (*Korolev and Nelson, 2011*).

Our work defines the expected pattern created by pair-wise interactions if the fitness effects of interactions are the main driving force of patterning. The generality of these ecological patterning conclusions awaits further experimental validations. Many simple communities grown in laboratory environments conform to these findings. For instance, competing bacterial species form a columnar or layered pattern (*Palmer et al., 2001*; *Kreth et al., 2005*; *An et al., 2006*; *Hallatschek et al., 2007*; *Bernstein et al., 2012*). *Burkholderia* sp. LB400 and *Pseudomonas* sp. B13(FR1) show spatial segregation as competitors when grown on citrate, but when grown on chlorobiphenyl they engage in metabolic commensalism and exhibit a layered pattern (*Nielsen et al., 2000*). Similarly, when grown on a non-selective carbon source, *Comamonas testosteroni* WDL7 outcompeted and covered *Variovorax* sp. WDL1, but when fed with an aromatic compound as the sole carbon source, competition switched to cooperation and the two populations intermixed (*Breugelmans et al., 2008*). Cooperative oral bacteria species intermixed when grown on saliva (*Palmer et al., 2001*; *Periasamy and Kolenbrander, 2009*). If cooperative intermixing is suspected, then examining whether the intermixing index increases linearly as a function of community height will be informative to exclude transient increases.

## Deviations from the expected ecological patterns suggest other major patterning forces

We have described the expected patterning between two cell populations if the fitness effects of interactions are the major driving force. These expectations are abstract in the sense that in reality, no interactions can exist in the absence of molecular mechanisms or an abiotic environment. Assigning expected patterns to different types of ecological interactions will allow us to identify deviations from expectations. Determining the causes of such deviations will help us better understand the biology of a community.

Fitness effects of ecological interactions rely on molecular mechanisms to manifest themselves. Different types of molecular mechanisms can conceivably alter patterning. For instance, in the fitness and the diffusion models and in the *S. cerevisiae* communities, cells divide upward when sufficiently confined horizontally. This type of cell rearrangement has also been observed in bacterial colonies (*Kreft et al., 1998*) and was adopted to model biofilm growth (*Xavier and Foster, 2007*). However, it is conceivable that if at least one population actively moves (*Houry et al., 2012*) or grows hyphae to penetrate the entire community, two populations might appear intermixed even if they do not cooperate. In biofilms of *Pseudomonas aeruginosa*, two populations differing only in fluorescent colors (*Klausen et al., 2003*) showed modest intermixing even though the expectation for equal-fitness competition is a columnar pattern with an intermixing index close to zero (*Figure 2D*, left). This modest intermixing was caused by *P. aeruginosa* differentiating into non-motile 'stalk' cells that anchored to the surface and motile cells that climbed up to form the mushroom-like caps (*Klausen et al., 2003*).

Environmental influences can also alter ecological patterning. For instance, if two cooperating populations have very different preferences for oxygen, then the two populations are likely not to intermix and instead form layers in which the aerobic population is exposed to oxygen while the anaerobic population is protected from oxygen.

## What can cause variability in patterning?

Variability in patterning has been observed within and between communities (*Christensen et al., 2002*; *Wilmes et al., 2008*; *Dekas et al., 2009*), even if they were grown in laboratory-controlled environments (*Christensen et al., 2002*). Stochastic events such as environmental variability, mutations, or fluctuations in initial conditions can all lead to variable patterns. For instance, if the straight-columnar pattern expected for equal-fitness competition is observed for the majority of community but a layered pattern is observed in occasional locations, then fitter mutants may have arisen in these locations, initiating unequal-fitness competition (*Hallatschek et al., 2007*; *Korolev et al., 2012*). For mutually antagonistic interactions, the fitness model showed that population ratios can quickly diverge. Thus, which population eventually dominates depends on the initial population ratio (*Figure 1—figure supplement 3*). In this case, stochastic variations in initial conditions can result in dramatically different patterns, giving rise to a phenomenon equivalent to 'survival of the first'. Indeed, different patterns have been observed for communities formed by two antagonistic bacteria species (*Kuramitsu et al., 2007*).

In summary, our work is conceptually analogous to that of the competitive exclusion principle (*Gause, 1934*). The competitive exclusion principle, also known as Gause's law, states that two species competing for the exact same resources cannot stably coexist. Analogous to how the competitive exclusion principle has created a framework to examine forces that cause species coexistence, our work on ecological patterning will hopefully lay the ground for examining mechanisms that shape patterning in microbial communities. We encourage comments, especially those pertaining to the generality of our conclusions, to be posted to the eLife website.

## Materials and methods

### Engineering yeast strains

In competitive communities, equal-fitness $G$ and $R$ strains were respectively WS931 (*MATa ste3::kanMX4 trp1-289::pRS404(TRP)-ADHp-venus-YFP*) and WS937 (*MATa ste3::kanMX4 trp1-289::pRS404(TRP)-ADHp-DsRed.T4*), both from the S288C background. For unequal-fitness competitive communities, $G$ was replaced by WS1246 (*MATa ho::loxP AMN1-BY Supercontig17(27163-27164)::TDH3p-yEGFP-loxP-kanMX-loxP*) from the RM11 background with a 20% fitness advantage over the S288C background. In commensal communities, $R^{\leftarrow L}_{\rightarrow A}$ and $G^{\leftarrow A}$ strains were respectively WS950 (*MATa ste3::kanMX4*

*lys2Δ0 ade4::ADE4(PUR6) trp1-289::pRS404(TRP)-ADHp-DsRed.T4*) and WS932 (*MATa ste3::kanMX4 ade8Δ0 trp1-289::pRS404(TRP)-ADHp-venus-YFP*), both from the S288C background. In cooperative communities, $R^{\leftarrow L}_{\rightarrow A}$ and $G^{\leftarrow A}_{\rightarrow L}$ strains were respectively WS950 and WS954 (*MATa ste3::kanMX4 ade8Δ0 lys21::LYS21(fbr) trp1-289::pRS404(TRP)-ADHp-venus-YFP*) from the S288C background.

## Culturing communities

For yeast communities, agarose columns were prepared by pouring SD minimal medium (*Sherman, 2002*) with 2% low melting temperature agarose in 1.3-ml deep-well plates (U96 DeepWell from Nunc, Penfield, NY), and after agarose had solidified, adding drops of melted SD agarose to make the surface flat. For commensal yeast communities, SD was supplemented with lysine (80 µM final concentration). Batch cultures of yeast strains were grown to exponential phase at 30°C in SD with supplements when necessary. Cells were washed free of supplements if any, mixed, and filtered on top of MF membrane (HAWP04700 from Millipore, Billerica, MA). Disks were cut from the membrane using a 6-mm-diameter Harris Uni-Core puncher and transferred to the top of agarose columns, unless otherwise stated.

For the coculture of *D. vulgaris* and *M. maripaludis*, a CDC reactor (BioSurface Technologies Corp., Bozeman, MT) was used for anaerobic biofilm growth. Biofilm coupon holders were modified to hold glass microscope slides (Fisher Scientific #12-544-1, Fisher Scientific, Pittsburgh, PA) cut to 7.6 × 1.8 cm. Cocultures were grown in CCM (*Walker et al., 2009*), a modified basal salt medium without choline chloride. Headspace was sparged with anoxic 80% $N_2$:20% $CO_2$, and the reactor was maintained at 30°C with stirring (150 rpm). The reactors were inoculated with planktonic coculture, and after cell attachment to the glass slides (48 hr), biofilms were allowed to develop and grow over time in the presence of planktonic cells, or initially drained of planktonic cells. Patterns in the two experimental regimes were similar.

## Flow cytometry

Flow cytometry was performed in a modified 4-laser FACS Calibur machine (DxP; CyTek Development, Fremont, CA). Cells in communities were diluted to ~$10^6$ cells/ml in $H_2O$. Each sample (90 µl) was supplemented with 10 µl of fluorescent bead stock (Thermo Scientific Fluoro-Max Cat# R0300 at ~8 × $10^6$/ml, counted using a Z2 Coulter counter and a hemacytometer) as a reference to determine total cell density and 3 µl of 1 µM ToPro3 (Invitrogen, Grand Island, NY) to mark dead cells. The laser and filter configurations for different fluorophores were 50 mW 488 nm laser with 530/30 filter for YFP, 75 mW 561 nm laser with 575/26 filter for DsRed, and 25 mW 639 nm laser with 660/20 filter for ToPro3. Using an automatic micro-sampling system (DxP; CyTek Development, Fremont, CA) samples in 96-well plates were processed for 60 s at a flow rate of 0.5–1 µl/s, recording $10^4$–$10^5$ events. FlowJo software (Tree Star, Ashland, OR) was used to measure the ratio of different populations of fluorescent cell against the bead standard in order to calculate cell densities.

## Fluorescence imaging

A Nikon TE2000 inverted microscope equipped with a Prior stage controller, a Sutter Lambda XL fluorescent lamp, and a Photometric SnapHQ CCD camera was controlled by custom LabVIEW software to auto-focus and acquire images. All images were taken at 10× magnification using a Nikon long working distance CFI Plan Fl objective (NA 0.30, WD 16). For imaging YFP- and DsRed-tagged strains, ET500/20×-ET535/30m-T515LP and ET545/30×-ET620/60m-T570LP filter cubes were used, respectively. Timelapse imaging took place in a 30°C chamber (In Vivo Scientific microscope incubator).

## Cryosectioning

To obtain vertical cross-sections of communities through cryosectioning, we froze communities in liquid nitrogen for 15 s and fixed them in methanol at −20°C. After 20 min, the communities were transferred to a pre-cooled empty plate at −20°C to allow methanol to evaporate, which typically took 4 hr. The communities were embedded in optimal cutting temperature (OCT) compound for 10 min at room temperature and subsequently frozen over dry ice and kept at −20°C for sectioning. We also froze down communities without fixing by directly embedding them in OCT (*Piccirillo et al., 2010*) and immediately freezing them on dry ice. Results obtained from the two procedures yielded similar conclusions.

For sectioning, communities embedded in OCT blocks were mounted on a cryotome. The blade was adjusted to ensure cross-sections of the community were vertical. For each community, typically

thirty to fifty 14-µm (3-cell-thick) sections were cut and transferred to glass slides. Cross-sections were imaged using the fluorescence microscope as described above. Only images of cross-sections that were minimally perturbed by the fixing and sectioning processes were included in the analysis. More details of the cryosectioning method can be found in (*Momeni and Shou, 2012*).

## Fluorescence in situ hybridization (FISH)

Biofilms of *D. vulgaris* and *M. maripaludis* were fixed in 4% paraformaldehyde for 4 hr, and then embedded in polyacrylamide (*Daims et al., 2006*). The embedded biofilms were dehydrated and hybridized in 1 ml buffer solution (0.9 M NaCl, 20 mM Tris–HCl (pH 8), 0.01% SDS, and 35% deionized formamide) with 3 ng each of probes EUB338 (GCT GCC TCC CGT AGG AGT) 5' and 3'-labeled with Cy3 and ARCH915 (GTG CTC CCC CGC CAA TTC CT) 5' and 3'-labeled with Cy5 (*Stoecker et al., 2010*) for 5 or 8 hr at 46°C in a humid chamber. Next, samples were washed in 50 ml prewarmed washing buffer (70 mM NaCl, 20 mM Tris–HCl pH 8, and 5 mM EDTA) at 47°C for 20 min, then dipped in ice cold ddH$_2$O and quickly dried with compressed air (*Amann et al., 1995*). Finally, each sample was mounted with Citifluor AF1 antifadent (Citiflour Ltd., Leicester, United Kingdom) and viewed using a Leica TCS SP5 II inverted confocal laser scanning microscope with 488, 561 and 633 nm lasers and appropriate filter sets for Cy3 and Cy5. Confocal voxel size was typically 0.24 × 0.24 × 0.49 µm.

## Spatial analysis

For both simulated and experimental sections, our unit of analysis was the size of one CCD camera field of view under a 10× objective which has a width of 0.7 mm. In any unit of analysis, community height is the value such that 90% of height values are below it. This choice is made to exclude artifacts such as height spikes in simulations. Images of experimental community sections were rotated in ImageJ such that the *x*–*z* axes of the frame matched those of the community. We further digitized these images into *f(x,z)* with assigned values of +1, −1, and 0 for pixels identified as population 1, population 2, and no-signal, respectively. No-signal pixels were defined as having a fluorescence intensity per unit exposure time less than 10–20% above the background in fluorescence channels. For the remaining pixels, the intensity values of 90th percentiles were found in each fluorescence channel. These values were used to normalize the corresponding green- or red-fluorescence intensity for each pixel whose identity was then assigned to be the color with the higher normalized value. In simulated communities with more than two populations, analysis was performed on two focal colors at a time, and in each analysis, pixels identified as other colors were treated as no-signal.

To compare levels of intermixing in different communities, we estimated the number of cell type transitions spanning community height. We define the intermixing index as the average number of color changes along community height: let $h(x_i)$ and $c(x_i)$ respectively be the local height and the number of color changes along *z* at the lateral position $x_i$. The intermixing index *IM* can be calculated as

$$IM = \left( \sum_{x_i} c(x_i) h(x_i) \right) \Big/ \left( \sum_{x_i} h(x_i) \right). \quad (1)$$

Intermixing is small for segregated patterns with few color changes along height, and increases when patches of different cell types successively appear on top of each other. Note that this choice of intermixing index yields small values when one population is very rare compared to the other. We weighted $c(x_i)$ by the local community height $h(x_i)$, thus giving more emphasis to taller regions. This is because taller regions have gone through more growth regulated by the fitness effects of interactions.

To estimate the vertical patch size $\lambda_z^*$, the ratio of height $h(x_i)$ to number of cell-type layers, $1 + c(x_i)$, was averaged across the community cross-section

$$\lambda_z^* = \left( \sum_{x_i} [h(x_i)]^2 / [1 + c(x_i)] \right) \Big/ \left( \sum_{x_i} h(x_i) \right). \quad (2)$$

For engineered yeast communities, typically 10–20 frames from different locations of each community were included in the analysis to ensure an unbiased representation of community patterns. For biofilms of *D. vulgaris* and *M. maripaludis*, vertical cross-sections ~2.4 µm apart were sampled from confocal z-stack images of biofilms.

## Calculating the characteristic patch size in cooperative yeast communities

The characteristic patch size can also be calculated as the effective length of interaction between two partners. We calculate how far inside the community released lysine can diffuse before being consumed; similar discussion applies to adenine. Assume one cell has released $\beta_L$ fmole of lysine that diffuses at most a distance $l$ before being consumed. We define the sphere of radius $l$ as the diffusion domain of the release event. The number of consuming cells within the diffusion domain can be estimated as $N_u = (2l/c)^3$, where $c$ is the diameter of a cell. The average time nutrient can diffuse in the diffusion domain before being consumed is $t_c = l^2/2D_1$, where $D_1 = 360$ µm²/s is the diffusion constant within the community. Assuming that each consuming cell takes up lysine with a rate $v \leq \alpha_L/T$, with $\alpha_L$ being the amount of lysine required for a cell division and $T$ being the minimum doubling time, we have

$$\beta_L = N_u v t_c \leq \left(\frac{2l}{c}\right)^3 \frac{\alpha_L}{T} \frac{l^2}{2D_1};$$

thus,

$$l \geq \left[\frac{\beta_L}{\alpha_L} \frac{D_1 T c^3}{4}\right]^{1/5} \approx 50\,\mu m.$$

Within radius $l$, the average number of release events within time $t_c$ is $(2l/c)^3 t_c d_G \approx 0.2$. Thus, the probability of a second release event occurring in the same diffusion domain before nutrient from the first release event has been consumed is low. Therefore, $l$ defines the interaction length scale. Note that diffusion constant contributes by fifth root to $l$, and therefore, different diffusion constants in the community should not considerably alter patterns, as observed in simulations (*Figure 3—figure supplement 1A*). The calculated $l$ of 50 µm is larger than the experimental patch size $\lambda_z^*$ of ~10 to 20 µm. What could account for this discrepancy? Experimentally measured diffusion constant of Sulforhodamine 101 in community (20 µm²/s) reduces $l$ to ~30 µm. In addition, cells may take up more nutrients than what is required for producing one daughter and may store extra nutrients in vacuoles (*Shou et al., 2007*), further reducing the estimated interaction length scale.

## The fitness model

The individual-based fitness model followed cell growth in a three-dimensional simulation grid of $100 \times 100 \times 300$ cells with periodic boundary conditions along the $x$ and $y$ directions. Consider population $i$ ($i = R$ or $G$) interacting with population $j$ ($j = G$ or $R$). Without loss of generality, consider a focal cell from population $i$. The growth of the focal cell is influenced by cells in its cubic three-axial interaction neighborhood (*Deutsch and Dormann, 2004*) defined by $l$-cell-width to the left, right, front, back, above, and below. Let $\varphi_i$ and $\varphi_j$ be the fraction occupancy of $i$ and $j$ in the interaction neighborhood, respectively. The growth rate of the focal cell is $r_i = [r_{i0} + r_{ij}\varphi_j(1-\varphi_i)][1-\chi(\varphi_i + \varphi_j)]$. $r_{i0}$ is the basal fitness (growth rate of $i$ without any interactions); $r_{ij}\varphi_j(1-\varphi_i)$ represents the fitness effect on $i$ by $j$, which increases with partner abundance and decreases with recipient abundance due to intra-population competition for partner; $[1-\chi(\varphi_i + \varphi_j)]$ reflects intra- and inter-population competition for shared resources with fitness decreasing as the neighborhood becomes more occupied. Cells were inoculated in the bottom surface of the simulation grid. In each simulation time step $\Delta t$, the probability of cell division is $r_i \Delta t$. A cell would divide either to the side if there was space within its $(x, y)$ planar confinement neighborhood of $n$-cell radius or upward otherwise (*Figure 1—figure supplement 1*). Parameters used are listed in *Figure 1—source data 1*. $\chi = 0.8$, $l = 3$, and $n = 2$ in all cases. See *Source code 1* for an example (and *Source code 4* for the MATLAB function).

## The diffusion model

The individual-based diffusion model followed actions of cells (nutrient uptake, cell division, cell death, and possibly release of nutrients) and the distribution of nutrients in a three-dimensional simulation grid. Since cell division and death occur at a time-scale much longer than diffusion and nutrient uptake, we used a multi-grid scheme in both space and time. In this model, a three-dimensional simulation domain consisted of cell grids representing individual cells and nutrient grids representing nutrient

concentrations (*Figure 2—figure supplement 2*). Cell actions and nutrient distributions were updated at discrete time steps over the simulation domain. Simulations were typically performed over an agarose domain of 0.75-mm length × 0.75-mm width × 24-mm depth and a community domain of 0.75-mm length × 0.75-mm width × 0.3-mm height with parameters listed in *Figure 2—source data 1*.

Nutrient concentrations as a function of space and time are based on the diffusion equation

$$\frac{\partial S}{\partial t} = \nabla \cdot (D\nabla S) - U + Q, \qquad (3)$$

with

$$U = v_m \frac{S}{S + K_{MM}} n_u \qquad (4)$$

$$Q = \rho n_q. \qquad (5)$$

*Equation (3)* states that $S$, the amount of limiting nutrient in a diffusion grid, depends on three processes: i) diffusion of nutrient with diffusion constant $D$, ii) uptake of nutrients (*Walther et al., 2005*) by cells ($U$), and iii) in cooperative communities, release of nutrients by the partner population ($Q$). In *equations (4) and (5)*, $n_u$ and $n_q$ are the number of consuming and releasing cells within the diffusion grid, respectively, $K_{MM}$ is the Michaelis-Menten constant for uptake, $v_m$ is the maximum uptake rate per cell, and $\rho$ is the release rate per cell. To solve this diffusion equation, we used a finite difference time–domain method (*Crank, 1980*) with no-flow ($\partial S / \partial z = 0$) boundary conditions applied to the top and bottom surfaces of the simulation domain and periodic boundary conditions applied to the four vertical sides of the domain.

Cell growth rate $r$ in the model is dictated by Monod's equation

$$r(S) = r_m \frac{S}{S + K_M}, \qquad (6)$$

in which $r_m$ is the maximum growth rate when nutrients are abundant, $S$ is the concentration of the limiting nutrient, and $K_M$ is the $S$ at which half maximal growth rate is achieved. We assume that individual cells take up nutrients with $K_{MM} = K_M$, and once they have accumulated the required amount of the limiting nutrient, cell division occurs.

To incorporate realistic assumptions about cell rearrangement upon division in the community, we monitored single yeast cells growing into microcolonies on solid media (*Figure 1—figure supplement 1A*). Initially, each cell budded in the same plane and pushed others in its immediate neighborhood to the side. Once a cell was completely surrounded on each side by roughly five cells, it budded upward. The same process was implemented in the diffusion model (*Figure 2—figure supplement 2*). It should be noted that by forcing the confined cells to bud strictly upward, the model underestimates intermixing. As a result, communities show more vertical features in simulations than experiments (*Figure 2F*).

Temporally, the update time-step for nutrient diffusion and uptake (~1 s) was smaller than that for cell division and death (~360 s). Spatially, cell actions took place on a cell grid with single-cell resolution (5 μm), while nutrient distributions were followed on a diffusion grid at lower spatial resolution (~15 to 60 μm). These values were selected considering the trade-off between simulation time and accuracy, while ensuring the stability of simulations. For instance, at each time-step within a diffusion grid, the total amount of nutrients consumed should be considerably smaller than available nutrients. In other words, assuming that the length of a diffusion grid is $nc$ where $c$ is the length of a cell grid which is equivalent to the size of a cell, there are at most $n^3$ cells in each diffusion grid and

$$S(nc)^3 > n^3 v_m \frac{S}{S + K_{MM}} dt_u, \qquad (7)$$

where $S$ is the concentration of the limiting nutrient in the spatial grid, $K_{MM}$ is the Michaelis-Menten coefficient for nutrient uptake, $v_m$ is the maximum uptake rate, and $dt_u$ is the uptake time-step. After simplifying *equation (7)*, we obtain

$$dt_u < \frac{S + K_{MM}}{v_m} c^3. \qquad (8)$$

**Table 1.** Summary of steady-state occupancy, conditions to achieve steady-state, and the stability of steady-state for six types of ecological interactions.

| Interaction | Steady-state occupancy | Steady-state condition | Stability |
|---|---|---|---|
| $G[\sim\sim]R$ | Any $\varphi_G$ | Only when $r_{G0} = r_{R0}$ | Unstable |
| $G[\uparrow\uparrow]R$ | $\varphi_{G^*} = \dfrac{r_{G0} - r_{R0} + r_{int}}{2r_{int}}$ | $r_{int} > \lvert r_{G0} - r_{R0} \rvert$ | Globally stable |
| $G[\sim\downarrow]R$ | $\varphi_{G^*} = \sqrt{\dfrac{r_{R0} - r_{G0}}{r_{int}}}$ | $r_{int} > r_{R0} - r_{G0} > 0$ | Unstable |
| $G[\sim\uparrow]R$ | $\varphi_{G^*} = \sqrt{\dfrac{r_{G0} - r_{R0}}{r_{int}}}$ | $r_{int} > r_{G0} - r_{R0} > 0$ | Globally stable |
| $G[\downarrow\downarrow]R$ | $\varphi_{G^*} = \dfrac{r_{R0} - r_{G0} + r_{int}}{2r_{int}}$ | $r_{int} > \lvert r_{G0} - r_{R0} \rvert$ | Unstable |
| $G[\downarrow\uparrow]R$ | $\varphi_{G^*} = \dfrac{1}{2} + \sqrt{\dfrac{r_{G0} - r_{R0}}{2r_{int}} - \dfrac{1}{4}}$ | $r_{int} > r_{G0} - r_{R0} > 0$<br>$r_{int} < 2(r_{G0} - r_{R0})$ | Locally stable (when $\varphi_G > \varphi_{G,c}$; see *Figure 1—figure supplement 2*) |

In the worst case scenario of $S$ being much smaller than $K_{MM}$, using parameters of our engineered yeast strains, we find $dt_u \sim 0.5$ s. Thus, we chose $dt_u = 0.5$–1 s as the time-step for updating the nutrient uptake and diffusion equations. To ensure stability of the finite-difference equations for diffusion, the diffusion grid-size ($nc$) and the time-step ($dt_u$) for the diffusion equation have to satisfy (*Iserles, 2009*)

$$dt_u < \frac{1}{2}\frac{(nc)^2}{D}, \qquad (9)$$

where $D$ is the diffusion constant in the region of interest. From this relation, we chose diffusion grid size $nc = 50$ μm, and consequently, each diffusion grid contains 10 × 10 × 10 cells (*Figure 2—figure supplement 2A*). Since diffusion is fast within each grid (~1 s), this choice of grid size is unlikely to introduce a notable error in our calculations. See *Source codes 2 and 3* for examples (and *Source code 4* for the MATLAB function).

## Requirements for steady-state ratios in the six types of communities

Let $\varphi_R$ and $\varphi_G$ be the fraction occupancy of $R$ and $G$ in an interaction neighborhood, respectively. $\varphi_{G^*}$ is the fraction occupancy of $G$ that leads to equal fitness of the two populations and thereby result in a steady state ratio within the interaction neighborhood. Following the assumptions of the fitness model, the growth rate of each population is $r_i = [r_{i0} + r_{ij}\varphi_j(1 - \varphi_i)][1 - \chi(\varphi_i + \varphi_j)]$ where $i = R$ or $G$ and $j = G$ or $R$. For simplicity, we assume that $\varphi_R + \varphi_G = 1$, which leads to $\hat{r}_i = r_i/(1 - \chi) = r_{i0} + r_{ij}\varphi_j^2$.

In two-population cooperative communities, using the simplifying assumption of $r_{RG} = r_{GR} = r_{int} > 0$, we have $\hat{r}_G - \hat{r}_R = (r_{G0} - r_{R0} + r_{int}) - 2r_{int}\varphi_G$ (*Figure 1—figure supplement 2B*). Setting $\hat{r}_G - \hat{r}_R$ to 0, the community can achieve a steady-state value of $\varphi_{G^*} = (r_{G0} - r_{R0} + r_{int})/2r_{int}$. To satisfy $0 < \varphi_{G^*} < 1$, $r_{int} > \lvert r_{G0} - r_{R0} \rvert$ which means that the interaction term has to be strong enough to overcome the difference in the basal fitness of the two populations. Here, $\varphi_{G^*}$ is stable, since $\hat{r}_G - \hat{r}_R$ is positive (favoring $G$) when $\varphi_G < \varphi_{G^*}$ and negative (favoring $R$) when $\varphi_G > \varphi_{G^*}$. At $\varphi_{G^*}$, two populations grow at the same rate and population ratio is fixed at $R:G = (1 - \varphi_{G^*}):\varphi_{G^*}$.

Similar analysis shows the existence of a stable partner ratio under commensalism $G[\sim\uparrow]R$. A steady-state ratio $\varphi_{G^*}$ can exist under exploitation. However, initial ratios below the critical value $\varphi_{G,c}$ (*Figure 1—figure supplement 2B*) will not converge to $\varphi_{G^*}$. Other interactions ([~ ~], [~ ↓], and [↓ ↓]) do not converge to a stable ratio. Similar conclusions on ratio convergence hold if we assume $\hat{r}_i = r_{i0} + r_{ij}f(\varphi_j)$ for any continuous function $f$ that monotonically increases with $\varphi_j$ (proof not shown). Results are summarized in *Table 1*.

## Acknowledgements

We would like to thank current and past Shou group members (Adam Waite, Chi-Chun Chen, David Skelding, Aric Capel, and Justin Burton), our Hutch colleagues (Harmit Malik, Daniel Gottschling, Suzannah Rutherford, Katie Peichel, Maulik Patel, Nitin Phadnis, Sarah Holte, William Hazelton, and Steve Andrews), our UW colleagues (Ben Kerr, David Stahl, Kristina Hillesland, Josephine Chandler, Ajai Dendekar, E. Peter Greenberg), Sri Ram, and Wallace Marshall for discussions. We are grateful to Kristina Hillesland for connecting the Shou and Fields groups.

## Additional information

### Funding

| Funder | Grant reference number | Author |
| --- | --- | --- |
| Gordon and Betty Moore Foundation | | Babak Momeni |
| Life Science Research Foundation | | Babak Momeni |
| National Science Foundation | DGE 0654336 | Kristen A Brileya |
| United States Department of Energy | DE-AC02-05CH11231 | Kristen A Brileya, Matthew W Fields |
| National Institutes of Health | DP2 OD006498-01 | Babak Momeni, Wenying Shou |
| W. M. Keck Foundation | | Wenying Shou |

The funders had no role in study design, data collection and interpretation, or the decision to submit the work for publication.

### Author contributions

BM, Conception and design, Acquisition of data, Analysis and interpretation of data, Drafting or revising the article; KAB, Conception and design, Acquisition of data, Drafting or revising the article; MWF, Conception and design, Drafting or revising the article; WS, Conception and design, Analysis and interpretation of data, Drafting or revising the article

## Additional files

**Source code 1.**
• Source code 1. MATLAB code for simulating cooperative communities in the fitness model.

**Source code 2.**
• Source code 2. MATLAB code for simulating cooperative communities in the diffusion model.

**Source code 3.**
• Source code 3. MATLAB code for simulating competitive communities in the diffusion model.

**Source code 4.**
• Source code 4. MATLAB function for in-plane rearrangement of cells in both the fitness and the diffusion models.

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
