## [Decision Letter]

Thank you for choosing to send your work entitled “Cooperation leads to partner intermixing in microbial communities” for consideration at *eLife*. Your article has been evaluated by a Senior Editor and 3 reviewers, one of whom is a member of our Board of Reviewing Editors.

The Reviewing Editor and the other reviewers discussed their comments before we reached this decision, and the Reviewing Editor has assembled the following comments based on the reviewers’ reports.

The study deals with a very interesting mixture of experimental, theoretical, and simulation approaches to arrive at basic rules of spatial patterning between obligatory cooperative partners in single-celled systems. The studies are based on a previously developed system of two yeast strains that can enter cooperative interactions. The present paper deals with the development of a spatial patterning model in conjunction with a series of experiments that use different starting conditions to infer the stability of the observed patterns. The results of these experiments are then compared with the respective simulation results to validate the theoretical approach. This in turn is taken to infer further, more complex interactions in multi-partner systems. An additional experimental validation comes from the use of two bacterial species that interact on biochemically very different principles.

The referees agree that the paper is very interesting, technically sound, and the whole work rests on a very rich database. The main research question addressed is certainly significant, and the authors elegantly investigate it by using a combination of experimental and theoretical approaches. There are no severe methodological problems that need to be addressed. However, there are some concerns about the generality of the findings. The current version reads in places as if general rules were derived; but, strictly speaking, they are only tested for systems with obligate cooperation. Also, the possibility of antagonistic interactions is not addressed at all. Still, it appears that the paper is a very strong starting point for future work into extended directions. Hence, we will by happy to offer publication in *eLife*, provided the following points are addressed in a revised version.

A) The authors should be very explicit about the limitations of their current study, as well as the strengths. This could be presented in an extended discussion.

B) One of the reviewers suggests additional experiments in this context, but after further discussions, we consider them to be optional:

“The authors investigate a very special case of cooperation, namely an obligate interaction. Both their synthetic yeast system and the Desulfovibrio–Methanococcus interaction represent a case of an obligate interaction: that is, none of the two partners can grow without the respective other. In this light, the finding that cooperative interactions mix more than other ecological interactions seems much less surprising. It may simply be the consequence of the properties that were previously built into the system by the experimenter (i.e., the obligate mutual dependency) rather than a pattern that generally emerges in cooperative interactions (which are not necessarily obligate in nature). Support the interpretation that the partner intermixing observed by the authors is limited to obligatory cooperation.

Another very prominent example of cooperation in a microbial community is the cooperative production of biofilms (e.g., Rainey & Rainey 2003 *Nature*). Here we would also see spatial patterns in a biofilm, but I doubt that one would find the enhanced population mixing (given that two biofilm-formers were differentially labeled) as shown by the authors.

A way to manipulate the degree of mutual dependency experimentally would be to add lysine and/or adenine to the medium that is used to grow the yeast strains (as has been done to mimic the commensalism). My expectations for such an experiment would be that with more ‘public goods’ added, the less tight the interaction gets and the less pronounced the population mixing will be. Such an outcome would further support the interpretation that the conclusions drawn do not apply to cooperative interactions within microbial communities in general, but be rather limited to obligate cooperative interactions.”

Please respond to this comment either by doing the experiment or by clearly discussing these points in the manuscript.

---

## [Author Response]

The current version reads in places as if general rules were derived; but, strictly speaking, they are only tested for systems with obligate cooperation. Also, the possibility of antagonistic interactions is not addressed at all.

When we originally wrote “general rules,” we meant general rules on patterning driven by the fitness effects of ecological interactions. We have modified “general rules” to “ecological patterning.” We have also eliminated the usage of “rules.” We tested facultative cooperation in the fitness model, and, unfortunately, experiments will require a significant body of work. Antagonistic interactions were tested in the fitness model. We have added a discussion about antagonistic interactions.

*A) The authors should be very explicit about the limitations of their current study, as well as the strengths. This could be presented in an extended discussion*.

We are not sure whether the “limitations” refer to the lack of experimental data for interactions, such as facultative cooperation, or the existence of other patterning forces that can cause deviations from the ecological patterning rules.

The former point is discussed below. We do not believe that the latter can be regarded as limitations of our study. There are different forces that shape patterns of communities, one of which is the fitness effects of ecological interactions between populations. We show that if such fitness effects are responsible for community patterns, strong cooperation is the only type of interaction that leads to population intermixing. To make an analogy, species coexistence is not considered a limitation of the competitive exclusion principle. There are other forces that make species not compete for exactly the same resources, and the competitive exclusion principle provides a framework of expectations that allows us to recognize and study those forces. We do agree that the original discussions on forces that cause deviations from the ecological patterning expectations could benefit from being more explicit and organized. Thus, we have expanded and reorganized discussion into three subsections.

*B) One of the reviewers suggests additional experiments in this context, but after further discussions, we consider them to be optional*:

*“The authors investigate a very special case of cooperation, namely an obligate interaction. Both their synthetic yeast system and the Desulfovibrio–Methanococcus interaction represent a case of an obligate interaction: that is, none of the two partners can grow without the respective other. In this light, the finding that cooperative interactions mix more than other ecological interactions seems much less surprising. It may simply be the consequence of the properties that were previously built into the system by the experimenter (i.e., the obligate mutual dependency) rather than a pattern that generally emerges in cooperative interactions (which are not necessarily obligate in nature). Seems to support the interpretation that the partner intermixing observed by the authors is limited to obligatory cooperation*.

The reviewer is correct in pointing out that all our experiments were performed on obligatory cooperative systems. We only considered facultative cooperation where we showed that the larger the fitness benefits are compared to the basal fitnesses of the two partners, the more quickly intermixing index increases as a function of community height. To make this point more explicit, we have added to the main text:

“Using the fitness model, we found that in facultative cooperation, small fitness benefits generated less intermixing than large fitness benefits. This result is intuitive: patterns of facultative cooperation should resemble those of obligatory cooperation (or competition) when benefits to both partners are large (or small) compared to the basal fitnesses of cooperative partners. In facultative cooperation with smaller fitness benefits, intermixing would be apparent if communities could grow to greater heights. Further experiments are required to test these predictions.”

In we showed that excessive benefits diminish the degree of intermixing. Excessive benefits are unrelated to whether cooperation is facultative or not. For example, if **A** completely depends on **B** for a metabolite, **B** can still supply this metabolite in excess.

*Another very prominent example of cooperation in a microbial community is the cooperative production of biofilms (e.g., Rainey & Rainey 2003 Nature). Here we would also see spatial patterns in a biofilm, but I doubt that one would find the enhanced population mixing (given that two biofilm-formers were differentially labeled) as shown by the authors*.

The cooperative production of biofilm (Rainey & Rainey, 2003) relied on cooperation among cells of the *same* population. This type of intra-population cooperation is different from the inter-population cooperation **A[**↑ ↑**]B** described here, in which the benefit produced by **A** is different from that produced by **B**. We have now made this point more explicit in the main text:

“…, while inter-population cooperation based on the exchange of distinct benefits would be **[**↑ ↑**]**.”

*A way to manipulate the degree of mutual dependency experimentally would be to add lysine and/or adenine to the medium that is used to grow the yeast strains (as has been done to mimic the commensalism). My expectations for such an experiment would be that with more ‘public goods’ added, the less tight the interaction gets and the less pronounced the population mixing will be. Such an outcome would further support the interpretation that the conclusions drawn do not apply to cooperative interactions within microbial communities in general, but be rather limited to obligate cooperative interactions.*”

We have considered the experimental scheme proposed by the reviewer. However, we realized that if we supplement adenine and lysine to the agarose, the cells would simply exhaust the supplements first (competition) and then the interaction would switch to obligatory cooperation in addition to competition for other shared resources. One could argue that a steady supply of a very low concentration of adenine and lysine would create a facultative cooperative system, but there is no easy way of evenly supplying all cells in the community. In addition, any flow system would also wash out some of the released metabolites. To create *bona fide* facultative cooperation, we would need to either screen for natural microbes for facultative cooperation or create reduction-of-function (rather than loss-of-function) alleles in enzymes mediating adenine and lysine biosynthesis, and ideally these mutations should also be non-revertible. This would call for two genetic screens for appropriate mutations, which is beyond the scope of our study. Our fitness model predicts that in facultative cooperation, if for both partners the fitness benefits are large compared to the basal fitness, intermixing will increase rapidly as a function of community height.